# Damage-tolerant material design motif derived from asymmetrical rotation

Wei Wang[1,2,4], Shu Jian Chen [2,4✉], Weiqiang Chen [3], Wenhui Duan [1✉], Jia Zie Lai[1] &
Kwesi Sagoe-Crentsil [1]

Motifs extracted from nature can lead to significant advances in materials design and have
been used to tackle the apparent exclusivity between strength and damage tolerance of
brittle materials. Here we present a segmental design motif found in arthropod exoskeleton,
in which asymmetrical rotational degree of freedom is used in damage control in contrast to
the conventional interfacial shear failure mechanism of existing design motifs. We realise this
design motif in a compression-resisting lightweight brittle material, demonstrating a unique
progressive failure behaviour that preserves material integrity with 60–80% of load-bearing
capacity at >50% of compressive strain. This rotational degree of freedom further enables a
periodic energy absorbance pattern during failure yielding 200% higher strength than the
corresponding cellular structure and up to 97.9% reduction of post-damage residual stress
compared with ductile materials. Fifty material combinations covering 27 types of materials
analysed display potential progressive failure behaviour by this design motif, thereby
establishing a broad spectrum of potential applications of the design motif for advanced
materials design, energy storage/conversion and architectural structures.

---

[1] Department of Civil Engineering, Monash University, Clayton, VIC 3800, Australia. [2] School of Civil Engineering, The University of Queensland, St Lucia, QLD 4072, Australia. [3] Department of Mechanical, Aerospace and Civil Engineering, School of Engineering, The University of Manchester, Manchester M13 9PL, United Kingdom. [4]These authors contributed equally: Wei Wang, Shu Jian Chen. ✉email: shujian.chen@uq.edu.au; wenhui.duan@monash.edu

Nature adopts various design motifs to design materials with optimised properties and functionalities using relatively inferior components. These design motifs are valuable sources of inspiration for modern materials design. Since the first discovery of the helical structure in 1972[1], there has been a drive to extract design motifs from more than 7 million living species in the world[2] to aid the fabrication of structured/structural materials. After almost 50 years of research, remarkable repetitions have been confirmed in most classes of species and only eight categories of design motifs have been extracted and adopted in materials design[3]. These motifs are commonly recognised as fibrous[4], helical[5], gradient[6], layered[7], tubular[8], cellular[9], suture[10] and overlapping[11]. The discovery and adoption of these design motifs have led to monumental advancement of structural materials in the fields of protection[12], construction[13] and energy storage and conversion[14].

A significant proportion of the design motifs are natural solutions to perceived mutual exclusivity that exists between strength and damage tolerance of brittle materials. One major mechanism used to achieve damage tolerance is to increase toughness by introducing a more tortuous crack path and interfacial shear. For example, the layered motif found in nacre[15] and tooth enamel[16] has been extensively investigated and is characterised by the introduction of interfaces within the primary structure to improve toughness through shear between layers[17,18]. The helical structure is another approach widely adopted in materials design to elongate the crack path by assembling layers of mineral fibres or fibrils at varying angles[19,20]. Lastly, the suture motif introduces wavy or interdigitating interfaces that interlock with frictional bonds to dissipate energy during elongation[21]. Despite the differences in geometry and assembly methods of the classified motifs, they share a similar mechanism for damage tolerance, namely, dissipating energy via shearing between brittle materials[21,22].

Here we report the mechanism and realisation of a design motif derived from the load-bearing segmental structure of arthropod exoskeleton systems. In the present study, we first elaborate on the concept and working mechanism of the design motif, followed by proof-of-concept and demonstration of its application in material design using a scalable approach. Load-bearing capacity, failure mode and extent of applicability of the motif's implementation are investigated and characterised via mechanical testing coupled with X-ray micro-computed tomography (micro-CT), microscopic morphological examination, quantitative nanomechanical mapping and large-scale discrete element method (DEM) simulations.

## Results

**Segmental design motif.** A material design motif is extracted from the segmental exoskeleton structure of the arthropod. This exoskeleton structure, which is found in more than 80% of known animal species[23] such as flea, scorpion, centipede, etc., is another way of combining high strength and damage tolerance. As shown in Fig. 1a, b, d, a typical arthropod leg comprises multiple segments, such as the tibia and tarsus, which are thin shells connected by rotation joints[24]. When the segments are compressed (Fig. 1c, e), such as during leaping, the segments rotate asymmetrically around the joints, which provides energy storage and releasing ability, as well as maximising load-bearing capacity[24,25]. Similar segmental structures are also found in other species serving similar functions, such as legs of mammals (e.g. human and bear), amphibians (e.g. frog and toad) and reptiles (e.g. turtle and lizard). These segmental structures all exhibit a superior load-bearing capacity and great energy absorption ability via asymmetrical rotation.

Compared with four of the known design motifs, which are effective in adapting damage, inclusive of the helical, suture, layered and overlapping structures, the segmental structure delivers damage tolerance via asymmetric segmental rotational (Fig. 1f). As shown in Fig. 1f, the helical, suture and layered structures target improved damage tolerance and toughness under tension. The overlapping structure is usually designed for high flexibility and puncture resistance[26–29], but is less effective for carrying the load in the longitudinal direction. The energy absorption ability is mainly achieved via the sliding between individual plates or scales[3,29,30]. By contrast, the segmental design motif is highly effective for load because rotation is restricted in one direction, which contributes to high-level compressive stress in the stronger brittle materials. Only when experiencing excessive deformation does the rotational degree of freedom allow the hard material to deform sideways to avoid crushing the hard material. The energy dissipated in the asymmetric rotation of a segment is governed by Eq. (1) (Fig. 1g):

$$U = \frac{1}{2}\mathbf{F}\cos\theta_t \cdot \delta_t + \int_0^{\theta_t} \mathbf{F}\sin\theta \cdot L \cdot d\theta \qquad (1)$$

where $\mathbf{F}$ is the force applied on a segment with length $L$; $\theta_t$ and $\delta_t$ are rotation and axial deformation at time $t$, respectively. It needs to be noted in this study we only considered the segmental structure uses asymmetrical rotation as the key energy absorption mechanism. The fish scale structures[26,31,32] and spine-like structures[33,34] relying on sliding and shifting as key energy absorption mechanisms, which may also be referred to as segmental structures in some literature, are not considered here. The asymmetrical rotation mechanism of the segmental design motif fills in the substantial gaps in the capability and functionality of the existing design motifs.

**Realisation and fabrication of segmental design motif.** With the understanding of the working principle of the segmental motif, the execution of such delicate joints in material design presents a key challenge, especially using available manufacturing methods. We examined, to the best of our knowledge, all feasible schemes to implement the segmental motif in material design and a scaffold-based surface coating scheme was developed to improve manufacturing efficiency to deliver large-scale fabrication. The fabrication scheme was inspired by the formation of exoskeletons in which the internal soft tissues serve as the platform to control the geometry and growth of the hard exoskeleton[22]. In our scheme, the design motif was considered as a composite of two constituents: the scaffold and coating materials. The scaffold, with a geometric design that allowed asymmetrical coatings, was first fabricated using a material with relatively low stiffness (Fig. 1h), then a fluid mixture containing stiffer aggregates and binder was dip-coated or spray-coated onto the scaffold (Fig. 1i–l). Then, via curing, hydration, mineralisation or ceramization, the fluid was hardened to form the final product.

Our results suggested that the scaffold and fluid properties must be tuned to achieve a delicate geometry that regulates deformation in both the soft scaffold and the hard shell to achieve rotation. When using a well-tuned scaffold and fluid system, the proposed scheme achieves much higher efficiency than other fabrication techniques that have been proposed to deliver such systems, including biomimetic mineralisation[35], freeze casting[36] and additive manufacturing[19], all of which are heavily limited by scale, the complexity of geometry and applicable materials (see Supplementary Table 1 for detailed comparison).

Our first successful demonstration of this design motif used the proposed fabrication scheme by combining a polymeric scaffold and a cementitious hard shell. The U-shape scaffold, illustrated in

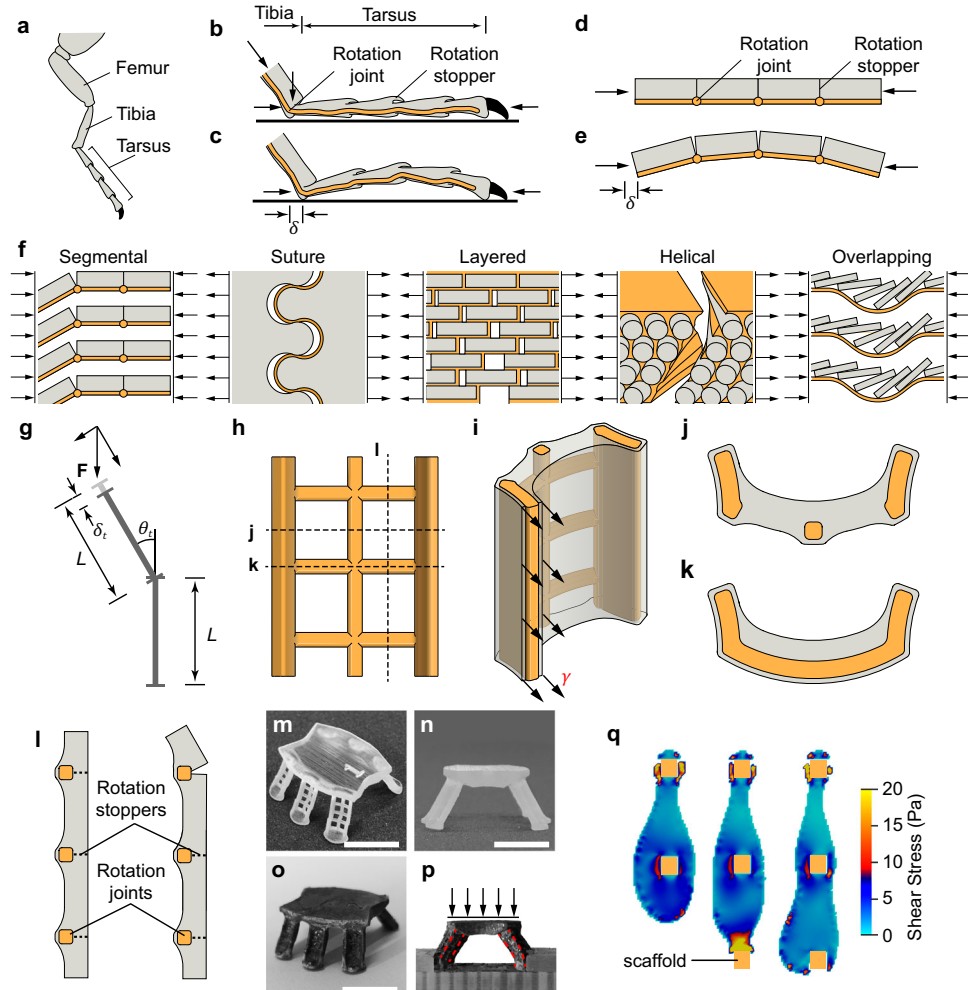

**Fig. 1 Segmental motif inspired by the structure of the arthropod exoskeleton. a** Segmental structure of arthropod leg. **b, c** Rotation of tarsal segments under compression. Grey indicates the stiff exoskeleton shell; yellow indicates soft tissue (tendons). **d, e** Kinematic model of tarsus under compression before and after segment rotation. Each design motif contains two types of material: stiff and strong material (grey) providing strength and soft material (yellow) improving toughness and flexibility. **f** Comparison of the inspired segmental motif with design motifs in the literature, including suture, layered, helical and overlapping motifs. A detailed comparison and discussion of the overlapping structure and segmental structure can be found in Supplementary Note 1. **g** Deformation and rotation of segment under loading. **h–l** Schematic of the realisation of the segmental motif. **h** Front view of the scaffold. The horizontal elements are denoted as beams and the vertical ones are denoted as columns. Effects of 3D printing orientations and thickness of the scaffold are presented in Supplementary Note 2. **i** Master view of the segmented scaffold after coating. The coated cement layer on the left side is cut and the surface tension is indicated by arrows. **j, k** Horizontal cross-sections showing asymmetrical coating. **l** Vertical cross-section showing asymmetrical coating before loading (left) and segment rotation after loading (right). The locations of the cross-sections are shown in (**h**) by the dotted lines. **m** Master view of the polymer scaffold of an artificial bug with a segmented polymer scaffold. **n** Front view of the polymer scaffold of the artificial bug. **o** Master view of the artificial bug after coating with cement paste. Scale bar = 1 cm. **p** Joint rotation of leg under compression. The artificial bug is supported and horizontal translation is constrained during loading. **q** Lattice Boltzmann method (LBM) simulation of the interaction between fluid and scaffolds under the co-effect of gravity and surface tension. The shear stress of the fluid at different time points is indicated by the colour map. Method of LBM simulation refers to Supplementary Note 3.

Fig. 1h, was specifically designed for two purposes: (1) to control the eccentricity of the coating (Fig. 1j, k) by the sidewalls of the U-shape (Fig. 1i) forming a rotation stopper in the clockwise direction (Fig. 1i) and (2) using a horizontal member (Fig. 1h, k) to introduce the flexibility of rotation in the counter-clockwise direction (Fig. 1l). Fresh Portland cement paste, the most consumed brittle/quasi-brittle human-made substance, was chosen as the coating material in the experimental trials. To ensure uniform and stable coating of the polymer scaffold, rheology properties, including the surface tension and yield stress, were carefully tuned and matched with the window size of the polymer scaffold. Initially, a proof-of-concept structural unit, resembling an insect with six supporting legs, was fabricated and

tested (Fig. 1m–p). That experiment showed that when the legs are loaded, the segments rotated counter-clockwise around the polymer horizontal member (Fig. 1p). The sway at the feet of the legs was prevented from lateral displacement similar to the working mechanism of the tarsus whereby a claw at the end is used to anchor to the surface to prevent lateral movement. Because clockwise rotation was stopped due to the eccentricity of the hard-mineral coating (Fig. 1l), the legs became very efficient in load-bearing. Our mechanical test results showed that the unit could easily sustain a load up to 10,000-fold of its own weight. It should be noted that the polymer scaffold contributed 1–3% of load-bearing capacity due to its very low elastic modulus and small volume fraction.

As described above, our proposed fabrication scheme provides a simple and effective way to execute the asymmetric joint and segmental structure. However, several key aspects must be understood to ensure the successful implementation of the scheme. In addition to the design feature of the scaffold illustrated in Fig. 1g, h, the rheological properties of the fluid are also critical to the dynamic formation of the coating. In contrast to Newtonian fluids[37], our results revealed that non-Newtonian fluids with Bingham behaviour contribute significantly to stabilisation of the coating. The lattice Boltzmann fluid simulation results revealed that when the fluid approached the polymer scaffold of the next tier, the shear stress and kinetic energy increased significantly due to the inherent surface tension of the binder (Fig. 1q). The dynamic forming process finishes when the kinematic energy of the paste fluid is dissipated by interparticle friction. The surface energy of the non-Newtonian fluid with air and the scaffold plays a key role in both the flow and spreading of the coating, but also contributes to the force equilibrium under gravity and interparticle friction when the coating is stabilised. Our calculation of the Gibbs free energy change during the dynamic forming process also suggested the rheological properties of the fluid (surface tension, $\gamma_L$, contact angle, $\theta$) should match with the geometry of the scaffold (void size, $\frac{b}{a}$) to contribute to uniform and stable film formation (for details, refer to Supplementary Note 4). The properties of the fluid can be tuned by varying the volume fraction of hard particles and the surface chemistry by surfactants (polycarbonate-based polymer was used in this case).

There is no doubt that the segmental structure demonstrated in Fig. 2 can be applied in materials design via a large variety of possible topological assembly methods such as 1D columns, 2D honeycombs, 3D lattices and 3D foams (Fig. 2a and Supplementary Fig. 1).

Here, we present a honeycomb assembly method, which helps exceed the performance limits of traditional brittle and quasi-brittle materials. 3D honeycomb structures can be formed using U-shape members arranged in a hexagonal manner as shown in Fig. 2a. The size of the U-shape can be varied to control the bulk density of the honeycombed material. (SH1 and SH2 in Fig. 2b–e). The assembly of the U-shape members was implemented in the scaffold design. As shown in Fig. 2b, d, the sidewalls of the U-shaped scaffold (Fig. 1f) merge with the windowed section of the adjacent units. This simplifies the scaffold design and reduces the use of polymeric material. The honeycombed material using the assembled scaffold can be fabricated via the coating of fluid (Fig. 2f). X-ray micro-CT scans (Fig. 2f–i) showed that an asymmetrical coating can be effectively formed in a similar fashion as on a U-shaped scaffold. After tuning both the scaffold design and the fluid properties, a thickness of the coating of 400–800 micrometres at the centre of the windows and 600–1000 micrometres at the level of the joints was achieved.

**Performance of segmental structure.** By applying compressive load while simultaneously observing real-time deformation patterns using X-ray micro-CT, both rational movements and crack propagation in the material were examined and shown in Fig. 3a–c. The micro-CT examination revealed that, before joint rotation, cracks are first initiated around the side of the scaffold with a thinner cover of hard shell, which was also confirmed in extended finite element method (XFEM) simulations (Supplementary Fig. 4). Next, the cracks propagated vertically into the thinner cover, which led to material spalling around the joint (Fig. 3a). After spalling occurred, the more flexible scaffold joint started to compress, leading the topmost segment to rotate towards the scaffold (counter-clockwise in Fig. 3c). As the rotation continued, the thicker hard shell on the other side changed

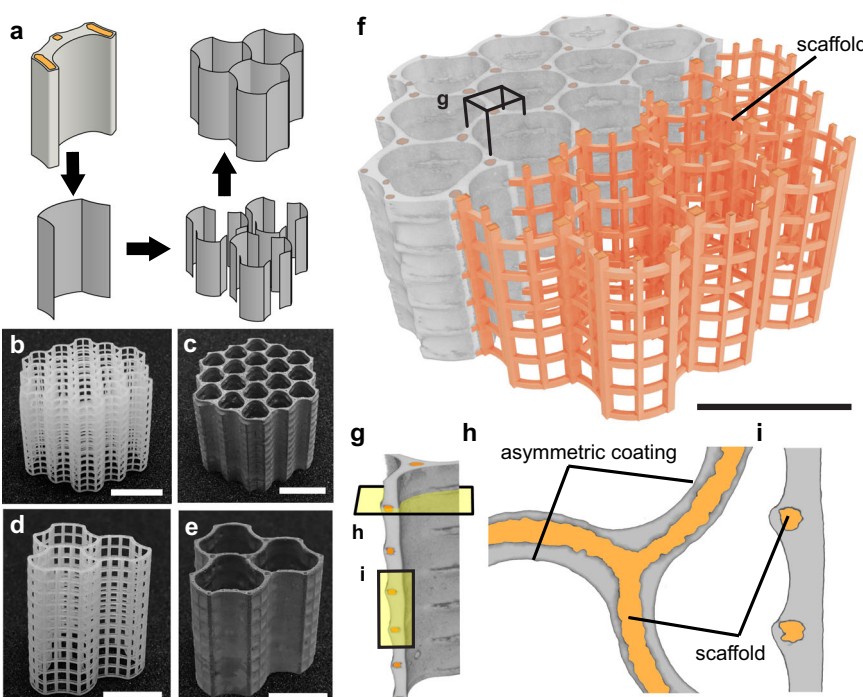

**Fig. 2 Assembly of segmented U-shape units into 2D honeycomb architecture. a** Schematic of the assembly steps. **b**, **c** Segmented honeycomb 1 (SH1) **b** before and **c** after coating. **d**, **e** Segmented honeycomb 2 (SH2) **d** before and **e** after coating (for more images and size details of SH1 and SH2 refer to Supplementary Fig. 2, Supplementary Fig. 3 and Supplementary Table 2). **f** X-ray micro-computed tomography (micro-CT) image showing cement coating of SH1. **g** Magnified view of the part outlined in (**f**). **h**, **i** Cross-sections shown in (**g**) show the asymmetric coating of cement paste on two sides of the polymer scaffold due to surface tension and polymer curvature. Scale bar = 1 cm.

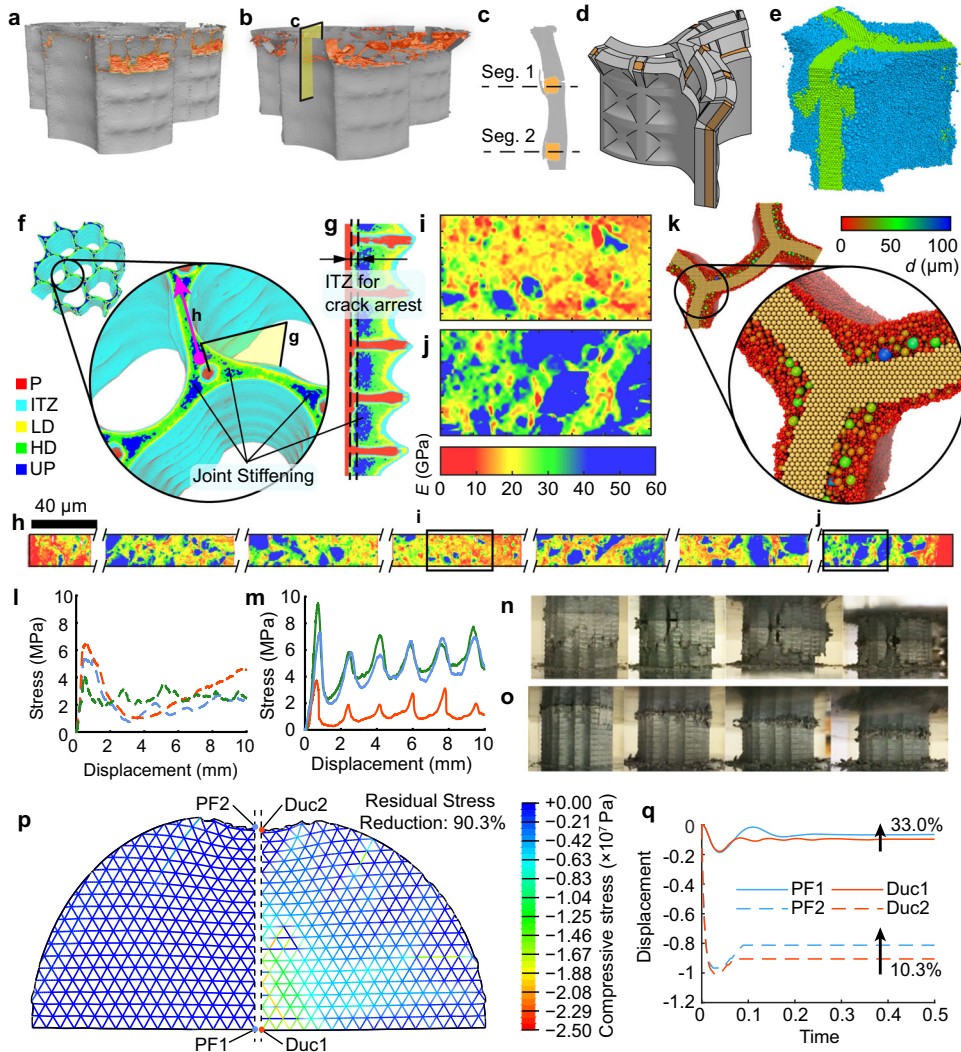

**Fig. 3 Failure behaviour of assembled honeycomb with segmental design motif. a, b** Typical X-ray micro-CT images obtained during the loading process of the segmented honeycomb (SH), with increasing deformation from loading state S1 (**a**) to loading state S2 (**b**). The undamaged structure is green-coloured, whereas the fracture surface is highlighted in orange. **c** Typical vertical cross-section showing rotation of the first segment around the polymer in (**b**). **d** Schematic of segment rotation of the assembled SH. **e** Discrete element method (DEM) simulation of the failure behaviour of SH. Details refer to Supplementary Note 5. **f** Typical X-ray micro-CT image showing the distribution of different phases within the SH. The Colour map indicates different phases within the SH, which are polymer (P), interfacial transition zone (ITZ), low-density hydration products (LD), high-density hydration products (HD) and unhydrated cement particles (UP). The determination of different phases refers to Supplementary Note 6. **g** Cross-section shown in (**f**). **h** Quantitative nanomechanical mapping (QNM) analysis indicates the elastic modulus (E) maps of the hardened SH. QNM scanning location and direction are indicated in (**e**). The distribution of E within the material is indicated by colour bar. More QNM results and the determination of different phases in QNM refer to Supplementary Note 7. **i, j** Magnified view of boxed regions in (**h**). **k** DEM simulation results show the distribution of the particles with different sizes in cement paste. The particle size is indicated by colour bar. **l, m** Typical loading curves showing the two different failure modes, comprising (**l**) irregular non-periodical progressive failure and (**m**) regular periodical progressive failure. **n, o** Typical pictures were taken during the loading process of the SH of **n** irregular non-periodical progress failure and **o** regular periodical progressive failure. **p** Finite element method (FEM) simulation of progressive failure (PF) material used as energy absorption material and compared with ductile (Duc) material. The Colour map indicates the distribution of residual stress (compressive stress) within the lattice structure. **q** Comparison of the deformation of the shell structure under impact. The locations under examination are indicated in (**p**). More FEM simulation results refer to Supplementary Note 8.

from compression to tension. Then, a horizontal crack formed that separated the first segment from the second (Fig. 3b, c). In addition, we also found the material in the top of the rotating segments was also being subjected to intensified fractures that contributed to greater energy absorbance (Supplementary Fig. 5). In the assembly structure, the asymmetric rotation was also found to be axisymmetric around the centre column of the scaffold where the three U-shapes connected with each other at 120-degree angles. A schematic showing the rotation mechanism of the segments in the assembled honeycombed material is

presented in Fig. 3d and a corresponding discrete element method simulation of the failure of the first tier after loading is shown in Fig. 3e.

We also discovered, via a combination of micro-CT, quantitative nanomechanical mapping (QNM) and discrete element method (DEM) simulation, that joint stiffening around the polymer elements also contributed to the initiation of joint rotation. We detected the formation of denser and stiffer materials around the polymer columns (Fig. 3f–k). Unbalanced force on the air–fluid interface[38] resulted in a net "squeezing"

force towards the fluid, which assisted the particles packing into a denser state and led to the concentration of unhydrated grains near the polymer columns (Fig. 3f–k). This joint stiffening effect created a stiffened ring with a higher nanoscale of Young's modulus around the polymer columns (Fig. 3f–k), which coupled with the interfacial transition zone between the polymer struts and hardened cement paste (Fig. 3g) to arrest the cracks[21] and guide them to propagate vertically within the segments (Fig. 3a and Supplementary Figs. 4, 5).

Our results also indicated that a balance between the element size of the scaffold and the thickness of the hard shell is crucial for joint rotation. The size of the horizontal scaffold members should be sufficient to arrest and guide cracks propagated horizontally to prevent propagation into adjacent segments (Fig. 3a and Supplementary Figs. 5, 6). It should be emphasised that the size of the polymer columns also needs to be tuned to ensure that all the segments in one tier can rotate simultaneously without pronounced crack propagation to the lower tier (Fig. 3d). We also found that improperly sized polymer elements will enable cracks to propagate through the whole specimen, resulting in an irregular non-periodical progressive failure (Supplementary Fig. 7 and Fig. 3l, n). We found that the strength of the scaffold needs to be around 1–3% of the segmented honeycomb to facilitate desired rotational behaviours.

Asymmetrically rotating segments allow the material to exhibit a unique periodic progressive failure mode under large deformation (Fig. 3m, o). The lower segments of the tuned segmented honeycomb remained intact and undamaged (Fig. 3o) even when loaded to >50% of engineering strain (50% of its initial length), whereas the segmented honeycomb without progressive failure fractured into several pieces (Fig. 3n). The honeycombed material can be classified as a type of metamaterial[39] due to its unique periodic progressive failure pattern and close to zero Poisson's ratio at large deformations. Based on the high regularity of progressive failure that can be achieved, it is not trivial to infer that the progressive failure can continue to further increase strain and deformation when used in taller structural compression-bearing members. The tier-wise progressive failure brings clear structural benefit over traditionally fabricated brittle materials, which can become highly unstable due to major cracks that often penetrate the whole structural member, resulting in the loss of integrity[40]. In a typical loading curve of the periodic progress failure process (Fig. 3l), multi peaks are observed with spacing around 1.75 mm, the height of each segment (Supplementary Table 2). Therefore, the intervals between the periodic peaks can be controlled via tuning of the scaffold and fluid for different segment sizes.

Periodic failure can also leverage the performance of quasi-brittle materials to match and even exceed ductile materials such as metals in a broad range of structural applications. As reported in the literature, progressive collapsing behaviour is a unique advantage of thin-walled ductile materials. Our results demonstrated that a quasi-brittle material can also demonstrate such behaviour. By considering progressive failure in the constitutional relationships of brittle and quasi-brittle materials, we conducted an examination of its effect as an energy absorption material embedded in typical hollow shells via FEM simulation. Compared with typical ductile materials, which show a constant strength after yielding[41], the progressive failure material shows a unique stress releasing mechanism. As shown in Fig. 3p, the residual stress of progressive failure material after local damage is significantly reduced by 90.5% (up to 97.9%, refer to the example in Supplementary Note 8) compared with ductile materials. In addition, as shown in Fig. 3q, the deformation due to local damage is also been reduced by up to 31.8% (up to 94.4%, refer to Supplementary Note 8).

The segmented honeycomb showed higher compressive strength than any other lightweight material using similar ingredients. The results of the mechanical test, together with several typical compressive strength tests of cellular foam concrete, are summarised in Fig. 4a. For example, foam concrete with a density around 250 kg/m$^3$ only has a strength of <1 MPa as reported in the literature[42,43]. By contrast, the assembled segmented honeycomb reported here had a strength of around 3 MPa, an increase of around 200%. As shown in Fig. 4a, we found the segmental design can reduce the degradation exponent $n$ between relative density and relative strength (see Supplementary Note 9) to 1.1–1.5, a vast improvement over the 2.0–3.4 of the traditional counterparts (for all referred literature see Supplementary Fig. 8). Besides, the material utilisation rate also improved from 59 to 79% using the segmental design (see Supplementary Note 9).

**Exploring of segmental structure in other materials**. We also explored the potential applications of the design motif in cementitious, ceramic, glass, metallic and polymetric materials. A list of typical materials that can potentially be implemented with the segmental design is presented in Fig. 4d, e which includes the commonly used (quasi-) brittle materials for coatings (e.g. cementitious material, ceramic, glass) and ductile materials for scaffold (e.g. polymer, metal). Although some of the materials may not be suitable for the fluid coating approach proposed here, the recent advances in 3D printing in metal[44,45] and ceramics[46–48] enabled the fabrication of these materials with complex geometry, such as the segmental structure proposed here.

The relationships of Young's modulus and compressive strength between polymer scaffold and cement shell are

$$\frac{E_{\text{Polymer}}}{E_{\text{Cement}}} \approx 0.1 \ \& \ \frac{\sigma_{\text{Polymer}}}{\sigma_{\text{Cement}}} \approx 0.78 \qquad (2)$$

Based on the relationships shown in Eq. (2), we predict the probabilities of progressive failure with the combinations of the two components for scaffold (S) and coatings (C) used are demonstrated in Fig. 4d. Here, we defined a progressive failure index (PFi) to estimate the probability for demonstrating a progressive failure, which is

$$\text{PFi} = \left| \log\left(\frac{E_{\text{S}}}{E_{\text{C}}}\right) - \log\left(\frac{E_{\text{Polymer}}}{E_{\text{Cement}}}\right) \right| + \left| \log\left(\frac{\sigma_{\text{S}}}{\sigma_{\text{C}}}\right) - \log\left(\frac{\sigma_{\text{Polymer}}}{\sigma_{\text{Cement}}}\right) \right| \qquad (3)$$

The probability of material combination showing a progressive failure behaviour is higher if the PFi is closer to 0. As shown in Fig. 4d, 50 combinations covering 27 materials show a *PFi* of less than 0.6 indicating a great potential for showing a progressive failure behaviour. In addition, 14 different materials covering cementitious, ceramic, glass, polymer and metallic materials were further highlighted in Fig. 4d. It can be seen that the combinations of polymer and cementitious material or ceramics (such as cement + PET/LDPE, boron nitride + PEI) show the highest probability of progressive failure behaviour due to the relative lower strength of polymer than ceramics and cementitious material. Besides, a metallic material combined with a high-strength ceramic, such as Ti6Al4V and silicon carbide, may also demonstrate a high potential of progressive failure behaviour. Considering the higher strength of metallic material than polymer, the segmental materials fabricated with the combination of metallic and ceramic materials will be more suitable for load-bearing purposes. Based on the very wide applicability demonstrated here, highly promising follow-up explorations are expected to discover a generation of material combinations and implementation methods of this design motif.

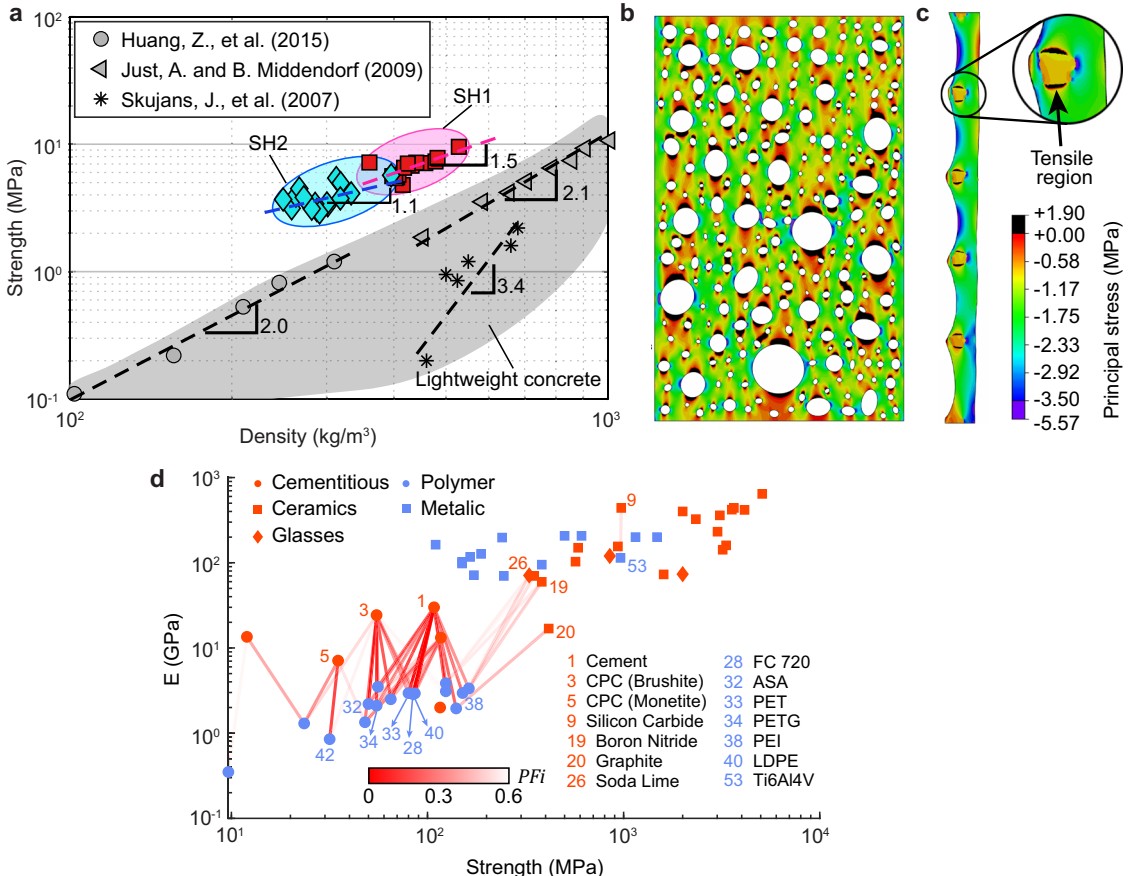

**Fig. 4 Compressive strength of segmented honeycomb and exploring in other materials. a** Graph comparing compressive strength of segmented honeycomb and lightweight cementitious material as reported in the literature. The grey region represents the compressive strength of lightweight concrete produced via traditional methods. The scatters indicate the compressive strength of lightweight concrete produced in three typical studies[43,50,51]. For all referred literature see Supplementary Fig. 8. Details of the test results of SH are summarised in Supplementary Table 3. **b** Results from extended finite element method (XFEM) simulation illustrating typical stress distribution of foam concrete when subjected to load from the top, with area shaded in black indicating tensile regions. Stress values are presented in the colour bar. XFEM simulation method refers to Supplementary Note 9. **c** Typical stress distribution of segmented honeycomb. **d** Young's modulus and compressive strength of materials adopted for coatings and scaffold. Lines indicate the combination of materials and line colours represents the progressive failure index (*PFi*) which indicates the probability of demonstrating progressive failure behaviour. Labels of scatters are material IDs in Supplementary Table 4 and Supplementary Table 5, which list all potential materials applicable for the segmental design motif.

In summary, we successfully extracted and applied the asymmetrical rotation-based segmental structures into material designs to fabricate lightweight composites exhibiting superior compressive strength and damage tolerance. We realised the application of the segmental structure in materials design. Previous research, including the investigation of fish scale structures[26,31,32] and spine-like structures[33,34], mainly focused on the structure level, such as the development of protective structures. The understanding of the application of segmental structure in material designs is still limited. The introduction of a segmental design motif fills a critical gap within known design motifs and opens up a broad spectrum of potential applicable brittle and quasi-brittle material. Our experimental results demonstrated successful realisation of the motif and its implementation in material design, resulting in metamaterials with which we demonstrated a unique progressive failure behaviour that preserves material integrity enabling 60–80% of the performance at >50% of compressive strain. The rotational degree of freedom also allowed a periodic energy absorbance pattern during failure with 100% more strength than traditional cellular foam and up to 97.9% of post-damage residual stress compared with ductile materials. We have also found the

segmental motif can be potentially implemented in various materials, among which fifty material combinations covering 27 types of different materials show great potential in achieving progressive failure behaviour, underpinning the vast potential application and benefits of the design motif.

## Methods

**Fabrication of scaffold**. The 3D scaffold was produced using an Objet Eden 360 3D polymer printer. An acrylic photopolymer was used as the printing material and another soluble acrylic photopolymer was used as the support material. NaOH solution with a concentration of 7% was used to clean the residue of the support material. The scaffold was designed as a combination of beam-column elements. Images of the 3D-printed polymer scaffold and CAD design are presented in Supplementary Figs. 2, 3. The thickness of the scaffold members was varied to produce scaffolds with different stiffness. Details of the cross-sectional thickness of the scaffolds and their corresponding stiffness are presented in Supplementary Table 2.

**Fabrication of U-shape units and segmented honeycomb**. General-purpose ordinary cement, with the chemical composition shown in Supplementary Table 6, was mixed with water (water to cement ratio: 0.25) using a high-shear mixer to produce a cementitious fluid. A standard mixing procedure from ASTM C1738/ C1738M-18[49] was adopted. The mini-slump test was conducted to measure the workability of the cement paste and a polycarboxylate-based superplasticiser (SP) was used to adjust the workability of the mixture at 125 ± 2 mm. After mixing, the

3D-printed polymer scaffold was immersed in the cementitious fluid for 3 s. While lifting the scaffold from the cementitious fluid, a coated segmented honeycomb was formed. The temperature and humidity of the environment during mixing and structure formation were controlled at $23 \pm 2\,°C$ and $50 \pm 2\%$. After structure formation, the segmented honeycomb was kept at $95 \pm 2\%$ humidity and $23 \pm 2\,°C$. After 16 h, the segmented honeycomb was further cured in saturated $Ca(OH)_2$ solution. After curing for 7 days, the hardened segmented honeycomb was dried at $60\,°C$ for 24 h and subsequently stored at $20 \pm 2\,°C$ in an ambient environment.

**Lattice Boltzmann method (LBM) simulation**. LBM simulation was carried out using open source software Palabos to simulate the interaction between the polymer scaffold and cementitious fluid. At the beginning of the simulation, fluid with properties similar to the fresh cement paste was allowed to drip onto the scaffold. The simulation was run until the stopping criteria were met. Different values of surface tension ($\gamma$) and Bingham yield stress ($\sigma$) of the fluid were used for the simulation, in which the combination that gave the most representative results in comparison with experimental results was chosen to be $\gamma = 0.01$ N/m and $\sigma = 11.99$ Pa (for details, see Supplementary Note 1).

**X-ray micro-computed tomography (micro-CT)**. In situ X-ray micro-CT was conducted before and during the compression test, using a ZEISS Xradia 520 Versa. The sample used for X-ray micro-CT had a height of $10 \pm 0.2$ mm. Approximately 1012 DICOM images were collected from each scan. The images were later reassembled into volume (grayscale value, GSV) data during the post-processing procedure.

**Quantitative nanomechanical mapping (QNM)**. Epoxy resin was used to mount and impregnate the hardened segmented honeycomb for sectioning and polishing. The sectioned samples were polished using a diamond with polypropylene glycol–ethanol mixture as a lubricant. Peak force QNM mapping was performed on the polished cross-section of the samples with a single crystal diamond tip (PDNISP-HS), using a Bruker Dimension Icon atomic force microscope (AFM). Before mapping commenced, a highly oriented pyrolytic graphite (HOPG) sample was used to calibrate the radius of the diamond tip.

**Extended finite element method (XFEM) simulation**. A 2D XFEM simulation was carried out using ABAQUS software to simulate crack propagation of the segmented honeycomb. Cross-sections of segmented honeycomb comprising multiple phases with different elastic moduli were modelled by the software. The crack initiation criterion was set to be 0.0001 tensile strain (for details, see Supplementary Note 9).

**Compression test**. Compression test was conducted using a universal loading machine Instron 4204 (50 kN). The samples were ground until their top and bottom surfaces were flat and parallel to the loading plate of the machine prior to testing. The samples were loaded at a rate of 0.1 mm/min in the first 10 min, under uniaxial loading. In the subsequent 5 min, the loading rate was slowly increased to 1 mm/min. An Instron Strain Gauge Extensometer 2630-112 was used to measure the deformation of the sample. The test stopped when the structure achieved 50% strain. Force–displacement data were obtained during the test.

## Data availability

The testing data generated in this study have been deposited in the figshare database under accession code: https://doi.org/10.6084/m9.figshare.18316532.v2.

## Code availability

The discrete element method (DEM) simulation code generated in this study have been deposited in the figshare database under accession code: https://doi.org/10.6084/m9.figshare.18316397.v2.

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

## Acknowledgements

The authors are grateful for the financial support of the Australian Research Council (IH150100006) in conducting this study. Dr. Shujian Chen is the recipient of an Australian Research Council Discovery Early Career Award (DE170100604) funded by the Australian Government. This work was performed in part at the Melbourne Centre for Nanofabrication (MCN) in the Victorian Node of the Australian National Fabrication Facility (ANFF). The authors also acknowledge the use of facilities within the Monash Centre for Electron Microscopy.

## Author contributions

W.W., S.J.C. and W.D conceived the study. W.W. and S.J.C. conducted the experiments and analysed the data. W.C. assisted the DEM simulation. J.Z.L. performed the XFEM simulation. W.W., S.J.C. and J.Z.L. wrote the manuscript. W.D. and K.S.-C. assisted in evaluating the results, discussion, and editing the manuscript. W.W. and S.J.C. contributed equally to this work.

## Competing interests

The authors declare no competing interests.

## Additional information

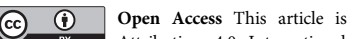

