## [Peer Review File · Nature Communications]

Title: Damage-tolerant material design motif derived from asymmetrical rotationREVIEWER COMMENTS

Reviewer #1 (Remarks to the Author):

The authors provide an experimental and theoretical exploration of the design motifs of arthropod exoskeletons. Based on a review of their manuscript the following revisions are provided:

1) The authors propose that the segmental structures found in the legs of arthropods could be considered as a new classification of design motifs similar to those previously described by Naleway et al. and have done an excellent job in identifying the structure and characterizing it in bioinspired materials. However, an important component of these design motifs inspired by nature is that they are found in a variety of species and are able to provide the same mechanical advantages regardless of their species host or material constituents (e.g., as the authors note: layered structures provide the same mechanical advantage when found fabricated from aragonite in sea shells and hydroxyapatite in human bone). The authors have only really identified these segmental structures as found in the legs of arthropods. Therefore, additional discussion on other, distinct species where this structure can be found would strengthen this manuscript.

2) The text generally compares these segmental design motifs to suture, layered, and helical design motifs. However, the design motif that seems to be the most similar is that of overlapping structures as, similar to the proposed segmental structures, these overlapping structures are able to absorb deformation through the motion and straining of distinct plates or rigid members. The authors argue that segmental structures are distinct from overlapping structures as segmental structures can only flex in one direction while overlapping structures can flex in multiple directions. However, this is not entirely true as overlapping structures, though providing flexural ability in many directions, tend to provide their defensive capabilities in one flexing mode (e.g., Vernerey and Barthelat, *Int. J. of Solids and Structures*, 2010 and Martini et al., *Acta Biomaterialia* 2017). While I generally do agree with the authors that segmental structures may be a distinct design motif, they should provide additional discussion to distinguish segmental structures and make a stronger case that they are not simply a subset of overlapping structures.

3) While I understand and generally applaud the authors in their intention to provide the reader with the mechanical properties of a litany of material combinations in segmental designs, Figure 4e is very difficult to understand. I might suggest either highlighting some important cases within the figure and discussing them in the text (e.g., the material combinations that would be best for biomedical materials, energy materials) or grouping the data in such a way that the reader can get the general gist without being overwhelmed by the information. In addition, the callout (on line 219) seems to be incorrect (for Figure 4d not Figure 4e) and additional discussion in the text could be beneficial.

Reviewer #2 (Remarks to the Author):

In this work, the authors introduce a new material motif that can increase strength and toughness in brittle materials. The main component in the new material motif is the introduction of a segmented architecture joined by compliant joints. Under compressive loads, the segmented structures are loaded under compression. Upon reaching a critical load, the rigid segments rotate at the joints, allowing a progressive failure behavior that increases the toughness compared with the constituent materials. The authors also present an innovative fabrication methodology to achieve these new materials and form 2D-Honeycombs. Although the results are interesting and of interest to the materials community, it is important to note that the topic of segmented materials is not new, and the work presented is incremental somehow incremental. It is suggested that the authors review the work in the area by Barthelat and co-workers, where the mechanics of these materials and how the increase in toughness and stiffness occurs is explained. Some of the relevant works are:

- Shafiei, A., Pro, J.W. and Barthelat, F., 2021. Bioinspired buckling of scaled skins. *Bioinspiration & Biomimetics*.
- Dalaq, A.S. and Barthelat, F., 2020. Manipulating the geometry of architected beams for maximum toughness and strength. *Materials & Design*, 194, p.108889.
- Dalaq, A.S. and Barthelat, F., 2019. Strength and stability in architected spine-like segmented structures. *International Journal of Solids and Structures*, 171, pp.146-157.
- Shafiei, A., Pro, J.W., Martini, R. and Barthelat, F., 2021. The very hard and the very soft: Modeling bio-inspired scaled skins using the discrete element method. *Journal of the Mechanics and Physics of Solids*, 146, p.104176.

Under this consideration, it is not clear what is the new “influence thinking” concept introduced by the authors. Therefore, this paper does not fit the scope to be published in the Nature Portfolio.

Responses to the Reviewer’s Comments

+ The authors are very grateful to the reviewers for their thorough, constructive and insightful comments as well as suggestions for improving the manuscript. Authors’ response and revisions made to the paper are summarized below.

Reviewer #1 Comments	Authors’ Response	Manuscript Amendment (highlighted in light blue)
1) The authors propose that the segmental structures found in the legs of arthropods could be considered as a new classification of design motifs similar to those previously described by Naleway et al. and have done an excellent job in identifying the structure and characterizing it in bioinspired materials. However, an important component of these design motifs inspired by nature is that they are found in a variety of species and are able to provide the same mechanical advantages regardless of their species host or material constituents (e.g., as the authors note: layered structures provide the same mechanical advantage when found fabricated from aragonite in sea shells and hydroxyapatite in human bone). The authors have only really identified these segmental structures as found in the legs of arthropods. Therefore, additional discussion on other, distinct species where this structure can be found would strengthen this manuscript.	Thanks for the reviewer’s comments. The segmental structures are composed of a number of stiff segments which are connected by asymmetric rotational joints. Such segmental structures not only exhibit superior strength to support the animal’s body but also provide flexibility to absorb energy during walking and jumping via asymmetric rotation [1, 2]. In our manuscript, we chose to use the exoskeleton structure of arthropod as a demonstration because the arthropod is the largest phylum in the animal kingdom. It contains 30-50 million species [3], such as flea, scorpion, centipede, and etc. These arthropods’ leg contains multiple segments which is also very similar to our designed segmented honeycomb’s structure. For example, an arthropod leg usually contains five segments which are coxa, trochanter, femur, tibia and tarsus [1, 2]. The tarsus may contain multiple subsegments [1, 2]. These segments rotate around joints to absorb/release energy during running or jumping [1]. In addition to arthropods, several other species also contain segmental structures which serve similar functions, such as legs of mammals (e.g. human, bear), amphibians (e.g. frog, toad) and reptiles (e.g. turtle, lizard). For example, human leg contains two main segments, femur and tibia, which will rotate around knee joint to absorb/release energy. To provide additional information, the manuscript is further revised to include the examples presented here.	Revised at line 55: This exoskeleton structure, which is found in more than 80% of known animal species [23] such as flea, scorpion, centipede, and etc, is another way of combining high strength and damage tolerance. Revised at line 61: Similar segmental structures are also found in other species serving similar functions, such as legs of mammals (e.g. human, bear), amphibians (e.g. frog, toad) and reptiles (e.g. turtle, lizard). These segmental structures all exhibit a superior load-bearing capacity and great energy absorption ability via asymmetrical rotation.
2) The text generally compares these segmental design motifs to suture, layered, and helical design motifs. However, the design motif that seems to be the most similar is that of overlapping structures as, similar to the proposed segmental structures, these overlapping structures are able to	Thanks for the reviewer’s comments. The key features of the overlapping structure have already been included in the original manuscript from line 62 to line 65. Here, more detailed comparison and discussion were added to distinguish the segmental structure from overlapping structure.	Revised at line 68: The overlapping structure is usually designed for high flexibility and puncture resistance [26-29], but is less effective for carrying load in the longitudinal direction. The energy absorption ability is mainly achieved

Responses to the Reviewer's Comments

absorb deformation through the motion and straining of distinct plates or rigid members. The authors argue that segmental structures are distinct from overlapping structures as segmental structures can only flex in one direction while overlapping structures can flex in multiple directions. However, this is not entirely true as overlapping structures, though providing flexural ability in many directions, tend to provide their defensive capabilities in one flexing mode (e.g., Vernerey and Barthelat, Int. J. of Solids and Structures, 2010 and Martini et al., Acta Biomaterialia 2017). While I generally do agree with the authors that segmental structures may be a distinct design motif, they should provide additional discussion to distinguish segmental structures and make a stronger case that they are not simply a subset of overlapping structures.	First of all, these two structures are designed for different loading conditions and serve for different functions. The segmental structure is usually loaded in the longitudinal direction. As we already explained and demonstrated in our original manuscript (line 48-51, line 201-209 and Fig.4a), the segmental structure shows a superior compressive strength in its longitudinal direction. Hence, the segmental structure can be adopted in animals' legs for load bearing purpose (support the self-weight). However, the overlapping structure is not effective to carry load in longitudinal direction. The overlapping structure (such as fish scales) is mainly designed for puncture resistance and the puncture force is in transverse direction [4-7]. Hence, the overlapping structure is usually used as an armour for protection purpose [4, 7-9]. Besides, the energy absorption mechanisms of these two structures are also different. The energy absorption ability of our segmental structure mainly depends on the asymmetrical rotation of the stiff segments around joints. For example, during the leaping of flea, the segments of flea's leg will rotate asymmetrically around joints to absorb and release energy [1]. However, for overlapping structure, the energy absorption of overlapping structure is achieved via the sliding between the individual plates or scales [7, 8, 10] (as shown in Supplementary Fig. 9). The literature also indicated individual plates/scales with architected and curved surface will increase the friction coefficient between the interface of plates or scales, leading to improved puncture resistance and energy absorption ability [11]. Besides, bending of the individual plates/scales [8] and brittle failure of the plates/scales [12] under high impact also assist the energy absorption. To provide additional explanation to readers, the manuscript is further revised and an addition supplementary note (Supplementary Note 1) is added. Fig 1f is also revised to further demonstrate the difference between these two structures.	via the sliding between individual plates or scales [3, 29, 30]. Inserted in Fig.1 caption: A detailed comparison and discussion of overlapping structure and segmental structure can be found in Supplementary Note 1. Inserted in Supplementary Information: Supplementary Note 1. Comparison between segmental and overlapping structures Both segmental and overlapping structures are consisted of multiple individual elements (such as segments, plates or scales). Here, a detailed comparison is given to distinguish the segmental structure from the overlapping structure. First of all, these two structures are designed for different loading conditions and serving for different functions. As shown in Supplementary Fig. 9a, the segmental structure is usually loaded and exhibits a superior compressive strength in its longitudinal direction. Hence, the segmental structure can be implemented in animals' legs for load bearing purpose (support the self-weight). However, the overlapping structure is not effective to carry load in longitudinal direction. The overlapping structure (such as fish scales) is mainly designed for puncture resistance and the puncture force is in transverse direction [26-29] as shown in Supplementary Fig. 9c. Hence, the overlapping structure is usually used as an armour for protection purposes [26, 29-31].  Supplementary Fig. 9. Comparison between segmental structure and overlapping structure. a, b The segmental structure before and after deformation. c, d The overlapping structure before and after deformation. Besides, the energy absorption mechanisms of these two structures are also different. The energy absorption ability of our segmental structure mainly depends on the asymmetrical rotation of the stiff segments around joints (Supplementary Fig. 9b). For example, during the leaping of flea, the
--	---	---

Responses to the Reviewer's Comments

		segments of flea's leg will rotate asymmetrically around joints to absorb and release energy [32]. However, for the overlapping structure, energy absorption is achieved via the sliding between the individual plates or scales [29, 30, 33] (Supplementary Fig. 9d). The literature also indicated the individual plates/scales with architected and curved surfaces increase the friction coefficient between the interface of plates or scales, leading to improved puncture resistance and energy absorption ability [34]. Besides, bending of the individual plates/scales [30] and brittle failure of the plates/scales [35] under high impact also assist with energy absorption capacity.
3) While I understand and generally applaud the authors in their intention to provide the reader with the mechanical properties of a litany of material combinations in segmental designs, Figure 4e is very difficult to understand. I might suggest either highlighting some important cases within the figure and discussing them in the text (e.g., the material combinations that would be best for biomedical materials, energy materials) or grouping the data in such a way that the reader can get the general gist without being overwhelmed by the information. In addition, the callout (on line 219) seems to be incorrect (for Figure 4d not Figure 4e) and additional discussion in the text could be beneficial.	Thanks for the reviewer's suggestions. The presentation of Fig 4 was modified and additional discussions were added as suggested. In the revised Fig 4d, we presented all of the materials in five groups which are cementitious material, ceramics and glasses for coating material, polymer and metallic material for scaffold, covering the most commonly used (quasi-) brittle and ductile materials. Although some of the materials may not be suitable for the fluid coating approach proposed here, recent advance of 3D printing in metal [13, 14] and ceramics [15-17] enabled the fabrication of these materials with complex geometry, such as the segmental structure proposed here. Based on the relationships of the Young's modulus and compressive strength between polymer scaffold and cement shell, we predict the probabilities of progressive failure with the combination of any two materials. The probabilities are indicated by the progressive failure index (PFI). As shown in Fig 4d, 50 combinations covering 27 different materials show a great potential in demonstrating the progressive failure behaviour. Several typical material combinations were highlight in Fig. 4d. It can be seen that the combinations of polymer and cementitious material or ceramics (such as cement + PET/LDPE, boron nitride + PEI) show the highest probability of progressive failure behaviour due to the relative lower strength of polymer than ceramics and	Revised at line 18 Fifty material combinations covering 27 types of materials analysed display potential progressive failure behaviour by this design motif, thereby establishing a broad spectrum of potential applications of the new motif for advanced materials design, energy storage/conversion and architectural structures. Revised at Fig. 4:  Revised at Fig. 4 Caption: Labels of scatters are material IDs in Supplementary Table 4 and Supplementary Table 5, which lists all potential materials applicable for the segmental design motif. Revised at line 222 We also explored the potential applications of the design motif in cementitious, ceramic, glass, metallic, and polymeric materials. A list of typical materials that can potentially be implemented with the segmental design is presented in Fig. 4d and e which include the commonly used (quasi-) brittle materials for coatings (e.g. cementitious material, ceramic, glass)

Responses to the Reviewer’s Comments

	cementitious material. Besides, a metallic material combined with a high strength ceramic, such as Ti6Al4V and silicon carbide, may also demonstrate a high potential of progressive failure behaviour. Considering the higher strength of metallic material than polymer, the segmental materials fabricated with the combination of metallic and ceramic materials will be more suitable for load bearing purposes. We have modified Fig 4d and removed Fig 4e. Only several important cases were highlighted in Fig 4d and the detailed list of all potential materials are summarized in Supplementary Table 4 and Supplementary Table 5. Additional discussion is also added.	and ductile materials for scaffold (e.g. polymer, metal). Although some of the materials may not be suitable for the fluid coating approach proposed here, recent advance of 3D printing in metal [46, 47] and ceramics [48-50] enabled the fabrication of these materials with complex geometry, such as the segmental structure proposed here. Revised at line 236: As shown in Fig. 4d, 50 combinations covering 27 materials show a PFI of less than 0.6 indicating a great potential for showing a progressive failure behaviour. In addition, 14 different materials covering cementitious, ceramic, glass, polymer and metallic materials were further highlighted in Error! Reference source not found.d. It can be seen that the combinations of polymer and cementitious material or ceramics (such as cement + PET/LDPE, boron nitride + PEI) show the highest probability of progressive failure behaviour due to the relative lower strength of polymer than ceramics and cementitious material. Besides, a metallic material combined with a high strength ceramic, such as Ti6Al4V and silicon carbide, may also demonstrate a high potential of progressive failure behaviour. Considering the higher strength of metallic material than polymer, the segmental materials fabricated with the combination of metallic and ceramic materials will be more suitable for load bearing purposes. Revised at line 260: We have also found the segmental motif can be potentially implemented in various materials, among which fifty material combinations covering 27 types of different materials show great potential in achieving progressive failure behaviour, underpinning the vast potential application and benefits of the new design motif. Inserted in Supplementary Table
--	--	--

Responses to the Reviewer’s Comments

		Supplementary Table 4. Properties of coating materials [22, 23]    No Material Young's Modulus (GPa) Compressive Strength (MPa)    1Cement30108 2PMMA bone cement2116 3Calcium phosphate cements (CPC)-brushite24.535 4Calcium phosphate cements (CPC)-apatite13.512 5Calcium phosphate cements (CPC)-monetite7.135 6Zinc Phosphate cement13.3117 7Sapphir4002000 8Cermet4403650 9Silicon Carbide440972.4 10Alumina3603095 11Aluminum Nitride3252335 12Silicon Nitride281.53012 13Mullite155.5935 14Zirconia142.153200 15Forsterite150586 16Cordierite70330 17Stearite103568.5 18Boron Carbide4174135 19Boron Nitride59.75382.5 20Graphite16.9115 21Titanium Diboride4203561.95 22Tungsten Carbide6455090 23Fused silica751600 24Silicon1603330 25Borosilicate73.52000 26Soda Lime71330 27Pyroceram120850   Supplementary Table 5. Properties of scaffold materials [24, 25]    No Material Young's Modulus (GPa) Compressive Strength (MPa)    28FullCure (FC) 7202.8784.3 29ABS2.0565 30Acrylic3.1124 31PLA3.0553.85 32ASA2.080 33PEI2.9580 34PEEK3.185 35PC1.95140 36PEEK3.855124 37PEKK3.35162.5 38PBI2.96151 39PP1.34448 40Nylon 6/62.9386.2 41LDPE0.259.65 42HDPE0.8531.5 43PTFE1.325.5 44Stainless Steel 316197.5240 45Stainless Steel 4202001480 46Copper127187.5 47Bronzes95382.5 48Brass117163 49Inconel 625207505 50Inconel 7182001150 51Aluminum (AIS10Mg)70245 52Titanium102.5150 53Ti6Al4V114964 54Cobalt207610 55Silver79.5172.5 56Platinum163110 57Gold98150  	No	Material	Young's Modulus (GPa)	Compressive Strength (MPa)	1	Cement	30	108	2	PMMA bone cement	2	116	3	Calcium phosphate cements (CPC)-brushite	24.5	35	4	Calcium phosphate cements (CPC)-apatite	13.5	12	5	Calcium phosphate cements (CPC)-monetite	7.1	35	6	Zinc Phosphate cement	13.3	117	7	Sapphir	400	2000	8	Cermet	440	3650	9	Silicon Carbide	440	972.4	10	Alumina	360	3095	11	Aluminum Nitride	325	2335	12	Silicon Nitride	281.5	3012	13	Mullite	155.5	935	14	Zirconia	142.15	3200	15	Forsterite	150	586	16	Cordierite	70	330	17	Stearite	103	568.5	18	Boron Carbide	417	4135	19	Boron Nitride	59.75	382.5	20	Graphite	16.9	115	21	Titanium Diboride	420	3561.95	22	Tungsten Carbide	645	5090	23	Fused silica	75	1600	24	Silicon	160	3330	25	Borosilicate	73.5	2000	26	Soda Lime	71	330	27	Pyroceram	120	850	No	Material	Young's Modulus (GPa)	Compressive Strength (MPa)	28	FullCure (FC) 720	2.87	84.3	29	ABS	2.05	65	30	Acrylic	3.1	124	31	PLA	3.05	53.85	32	ASA	2.0	80	33	PEI	2.95	80	34	PEEK	3.1	85	35	PC	1.95	140	36	PEEK	3.855	124	37	PEKK	3.35	162.5	38	PBI	2.96	151	39	PP	1.344	48	40	Nylon 6/6	2.93	86.2	41	LDPE	0.25	9.65	42	HDPE	0.85	31.5	43	PTFE	1.3	25.5	44	Stainless Steel 316	197.5	240	45	Stainless Steel 420	200	1480	46	Copper	127	187.5	47	Bronzes	95	382.5	48	Brass	117	163	49	Inconel 625	207	505	50	Inconel 718	200	1150	51	Aluminum (AIS10Mg)	70	245	52	Titanium	102.5	150	53	Ti6Al4V	114	964	54	Cobalt	207	610	55	Silver	79.5	172.5	56	Platinum	163	110	57	Gold	98	150
No	Material	Young's Modulus (GPa)	Compressive Strength (MPa)																																																																																																																																																																																																																																											
1	Cement	30	108																																																																																																																																																																																																																																											
2	PMMA bone cement	2	116																																																																																																																																																																																																																																											
3	Calcium phosphate cements (CPC)-brushite	24.5	35																																																																																																																																																																																																																																											
4	Calcium phosphate cements (CPC)-apatite	13.5	12																																																																																																																																																																																																																																											
5	Calcium phosphate cements (CPC)-monetite	7.1	35																																																																																																																																																																																																																																											
6	Zinc Phosphate cement	13.3	117																																																																																																																																																																																																																																											
7	Sapphir	400	2000																																																																																																																																																																																																																																											
8	Cermet	440	3650																																																																																																																																																																																																																																											
9	Silicon Carbide	440	972.4																																																																																																																																																																																																																																											
10	Alumina	360	3095																																																																																																																																																																																																																																											
11	Aluminum Nitride	325	2335																																																																																																																																																																																																																																											
12	Silicon Nitride	281.5	3012																																																																																																																																																																																																																																											
13	Mullite	155.5	935																																																																																																																																																																																																																																											
14	Zirconia	142.15	3200																																																																																																																																																																																																																																											
15	Forsterite	150	586																																																																																																																																																																																																																																											
16	Cordierite	70	330																																																																																																																																																																																																																																											
17	Stearite	103	568.5																																																																																																																																																																																																																																											
18	Boron Carbide	417	4135																																																																																																																																																																																																																																											
19	Boron Nitride	59.75	382.5																																																																																																																																																																																																																																											
20	Graphite	16.9	115																																																																																																																																																																																																																																											
21	Titanium Diboride	420	3561.95																																																																																																																																																																																																																																											
22	Tungsten Carbide	645	5090																																																																																																																																																																																																																																											
23	Fused silica	75	1600																																																																																																																																																																																																																																											
24	Silicon	160	3330																																																																																																																																																																																																																																											
25	Borosilicate	73.5	2000																																																																																																																																																																																																																																											
26	Soda Lime	71	330																																																																																																																																																																																																																																											
27	Pyroceram	120	850																																																																																																																																																																																																																																											
No	Material	Young's Modulus (GPa)	Compressive Strength (MPa)																																																																																																																																																																																																																																											
28	FullCure (FC) 720	2.87	84.3																																																																																																																																																																																																																																											
29	ABS	2.05	65																																																																																																																																																																																																																																											
30	Acrylic	3.1	124																																																																																																																																																																																																																																											
31	PLA	3.05	53.85																																																																																																																																																																																																																																											
32	ASA	2.0	80																																																																																																																																																																																																																																											
33	PEI	2.95	80																																																																																																																																																																																																																																											
34	PEEK	3.1	85																																																																																																																																																																																																																																											
35	PC	1.95	140																																																																																																																																																																																																																																											
36	PEEK	3.855	124																																																																																																																																																																																																																																											
37	PEKK	3.35	162.5																																																																																																																																																																																																																																											
38	PBI	2.96	151																																																																																																																																																																																																																																											
39	PP	1.344	48																																																																																																																																																																																																																																											
40	Nylon 6/6	2.93	86.2																																																																																																																																																																																																																																											
41	LDPE	0.25	9.65																																																																																																																																																																																																																																											
42	HDPE	0.85	31.5																																																																																																																																																																																																																																											
43	PTFE	1.3	25.5																																																																																																																																																																																																																																											
44	Stainless Steel 316	197.5	240																																																																																																																																																																																																																																											
45	Stainless Steel 420	200	1480																																																																																																																																																																																																																																											
46	Copper	127	187.5																																																																																																																																																																																																																																											
47	Bronzes	95	382.5																																																																																																																																																																																																																																											
48	Brass	117	163																																																																																																																																																																																																																																											
49	Inconel 625	207	505																																																																																																																																																																																																																																											
50	Inconel 718	200	1150																																																																																																																																																																																																																																											
51	Aluminum (AIS10Mg)	70	245																																																																																																																																																																																																																																											
52	Titanium	102.5	150																																																																																																																																																																																																																																											
53	Ti6Al4V	114	964																																																																																																																																																																																																																																											
54	Cobalt	207	610																																																																																																																																																																																																																																											
55	Silver	79.5	172.5																																																																																																																																																																																																																																											
56	Platinum	163	110																																																																																																																																																																																																																																											
57	Gold	98	150																																																																																																																																																																																																																																											
Reviewer #2 Comments In this work, the authors introduce a new material motif that can increase strength and toughness in brittle materials. The main component in the new material motif is the introduction of a segmented architecture joined by compliant joints. Under compressive loads, the segmented structures are loaded under compression. Upon reaching a critical load, the rigid segments rotate at the joints, allowing a progressive failure behavior that increases the toughness compared with the constituent materials. The authors also present an innovative fabrication methodology to achieve these new materials and form 2D-Honeycombs. Although the results are interesting and of interest to the materials community, it is important to note that the topic of segmented materials is not new, and the work presented is incremental somehow incremental. It is suggested	Thanks for the reviewer’s comments. In our manuscript, the segmental structure is referred to the structure which adopts asymmetrical rotation as the key energy absorption mechanisms. However, the segmental structure in literature is defined as a structure consisting of or divided into multiple segments, plates or scales, such as the fish scale structures [4, 18, 19] and spine-like structures [11, 20]. In the literature (including the literature the reviewer listed here [4, 11, 18, 20]), the researchers mainly focused on the fish scale structures [4, 18] and spine-like structures [11, 20], where sliding and interlocking of individual elements are the main energy absorption mechanisms [4, 11]. However, as we already explained in Reviewer 1’s comment #2, the energy absorption	Revised at line 77: It needs to be noted in this study we only considered that the segmental structure uses asymmetrical rotation as the key energy absorption mechanism. The fish scale structures [26, 31, 32] and spine-like structures [33, 34] relying on sliding and shifting as key energy absorption mechanisms, which may also be broadly referred to as segmental structures in some literature, are not considered here. The asymmetrical rotation mechanism of segmental design motif fills in the substantial gap in capability and functionality of the existing design motifs. Revised at line 248: In summary, we successfully extracted and applied the asymmetrical rotation-based segmental structures into material designs to fabricate																																																																																																																																																																																																																																												

Responses to the Reviewer’s Comments

that the authors review the work in the area by Barthelat and co-workers, where the mechanics of these materials and how the increase in toughness and stiffness occurs is explained. Some of the relevant works are:  • Shafiei, A., Pro, J.W. and Barthelat, F., 2021. Bioinspired buckling of scaled skins. Bioinspiration & Biomimetics. • Dalaq, A.S. and Barthelat, F., 2020. Manipulating the geometry of architected beams for maximum toughness and strength. Materials & Design, 194, p.108889. • Dalaq, A.S. and Barthelat, F., 2019. Strength and stability in architected spine-like segmented structures. International Journal of Solids and Structures, 171, pp.146-157. • Shafiei, A., Pro, J.W., Martini, R. and Barthelat, F., 2021. The very hard and the very soft: Modeling bio-inspired scaled skins using the discrete element method. Journal of the Mechanics and Physics of Solids, 146, p.104176. Under this consideration, it is not clear what is the new “influence thinking” concept introduced by the authors. Therefore, this paper does not fit the scope to be published in the Nature Portfolio	mechanism of the segmental structure investigated here is asymmetrical rotation, which is fundamentally different from the sliding mechanism. Current published understanding of the asymmetrical rotation of the segmental structure and its energy absorption capacity is still limited. The new “influence thinking” concept introduced by this manuscript is that we, for the first time, extracted and applied the asymmetrical rotation-based segmental structures into material designs to fabricate lightweight composites exhibiting superior compressive strength and damage tolerance. This is the first realization of the application of the segmental structure in materials design. Previous researches, including the investigation of fish scale structures [4, 18, 19] and spine-like structures [11, 20], mainly focused on the structure level, such as the developing of protective structures. In addition, we also proposed a novel fabrication process utilising 3D printing and fluid assisted coating. This segmental design motif and the proposed fabrication approach can also be applied to a wide range of materials to fabricate new lightweight, strong and tough composites. To provide further explanation on the new concept and distinguish the segmental structure in literature, the manuscript is further revised to include the details explained here.	lightweight composites exhibiting superior compressive strength and damage tolerance. This is the first realization of the application of the segmental structure in materials design. Previous researches, including the investigation of fish scale structures [26, 31, 32] and spine-like structures [33, 34], mainly focused on the structure level, such as the developing of protective structures. Understandings of the application of segmental structure in material designs is still limited.
--	---	---

Responses to the Reviewer's Comments

References

1. Sutton, G.P. and M. Burrows, *Biomechanics of jumping in the flea*. Journal of Experimental Biology, 2011. **214**(5): p. 836-847.
2. Albright, S., E. Fichter, and B. Fichter, *Kinematic model for arthropod legs and other manipulators*. 1994.
3. Adis, J., *Thirty million arthropod species—too many or too few?* Journal of Tropical Ecology, 1990. **6**(1): p. 115-118.
4. Shafiei, A., et al., *The very hard and the very soft: Modeling bio-inspired scaled skins using the discrete element method*. Journal of the Mechanics and Physics of Solids, 2021. **146**: p. 104176.
5. Ghods, S., et al., *Designed for resistance to puncture: the dynamic response of fish scales*. Journal of the mechanical behavior of biomedical materials, 2019. **90**: p. 451-459.
6. Zhu, D., et al., *Structure and mechanical performance of a “modern” fish scale*. Advanced Engineering Materials, 2012. **14**(4): p. B185-B194.
7. Martini, R., Y. Balit, and F. Barthelat, *A comparative study of bio-inspired protective scales using 3D printing and mechanical testing*. Acta biomaterialia, 2017. **55**: p. 360-372.
8. Vernerey, F.J. and F. Barthelat, *On the mechanics of fishscale structures*. International Journal of Solids and Structures, 2010. **47**(17): p. 2268-2275.
9. Zhu, D., et al., *Puncture resistance of the scaled skin from striped bass: Collective mechanisms and inspiration for new flexible armor designs*. Journal of the Mechanical Behavior of Biomedical Materials, 2013. **24**: p. 30-40.
10. Naleway, S.E., et al., *Structural design elements in biological materials: application to bioinspiration*. Advanced materials, 2015. **27**(37): p. 5455-5476.
11. Dalaq, A.S. and F. Barthelat, *Manipulating the geometry of architected beams for maximum toughness and strength*. Materials & Design, 2020. **194**: p. 108889.
12. Liu, P., et al., *Numerical simulation of ballistic impact behavior of bio-inspired scale-like protection system*. Materials & Design, 2016. **99**: p. 201-210.
13. Buchanan, C. and L. Gardner, *Metal 3D printing in construction: A review of methods, research, applications, opportunities and challenges*. Engineering Structures, 2019. **180**: p. 332-348.
14. Duda, T. and L.V. Raghavan, *3D metal printing technology*. IFAC-PapersOnLine, 2016. **49**(29): p. 103-110.
15. Owen, D., et al., *3D printing of ceramic components using a customized 3D ceramic printer*. Progress in additive manufacturing, 2018. **3**(1): p. 3-9.
16. Du, X., S. Fu, and Y. Zhu, *3D printing of ceramic-based scaffolds for bone tissue engineering: an overview*. Journal of Materials Chemistry B, 2018. **6**(27): p. 4397-4412.
17. Li, F., et al., *Digital light processing 3D printing of ceramic shell for precision casting*. Materials Letters, 2020. **276**: p. 128037.
18. Shafiei, A., J.W. Pro, and F. Barthelat, *Bioinspired buckling of scaled skins*. Bioinspiration & Biomimetics, 2021.
19. Mirkhalaf, M., et al., *Toughness by segmentation: Fabrication, testing and micromechanics of architected ceramic panels for impact applications*. International Journal of Solids and Structures, 2019. **158**: p. 52-65.
20. Dalaq, A.S. and F. Barthelat, *Strength and stability in architected spine-like segmented structures*. International Journal of Solids and Structures, 2019. **171**: p. 146-157.

REVIEWERS' COMMENTS

Reviewer #1 (Remarks to the Author):

The authors have addressed my concerns and I believe that the manuscript should be accepted.

Reviewer #2 (Remarks to the Author):

Thanks to the authors for addressing the comments and clarifying the differences of the new architecture with other segmented materials.

With the new edits in the manuscript, it is now clear the differences and the novelty of this Asymmetric Rotation design. I recommend this work for publication in the current state.